

# The ability of the green peach aphid (*Myzus persicae*) to penetrate mesh crop covers used to protect potato crops against tomato potato psyllid (*Bactericera cockerelli*)

Howard London[1,2], David J. Saville[1,3], Charles N. Merfield[4], Oluwashola Olaniyan[1] and Stephen D. Wratten[1]

[1] Bio-Protection Research Centre, Lincoln University, Lincoln, New Zealand
[2] National Agricultural Research and Extension Institute, Mon Repos, East Coast Demerara, Guyana
[3] Saville Statistical Consulting Limited, Lincoln, Canterbury, New Zealand
[4] The BHU Future Farming Centre, Lincoln University, Lincoln, New Zealand

Corresponding author
Charles N. Merfield,
charles.merfield@bhu.org.nz

## ABSTRACT

In Central and North America, Australia and New Zealand, potato (*Solanum tuberosum*) crops are attacked by *Bactericera cockerelli*, the tomato potato psyllid (TPP). 'Mesh crop covers' which are used in Europe and Israel to protect crops from insect pests, have been used experimentally in New Zealand for TPP control. While the covers have been effective for TPP management, the green peach aphid (GPA, *Myzus persicae*) has been found in large numbers under the mesh crop covers. This study investigated the ability of the GPA to penetrate different mesh hole sizes. Experiments using four sizes (0.15 × 0.15, 0.15 × 0.35, 0.3 × 0.3 and 0.6 × 0.6 mm) were carried out under laboratory conditions to investigate: (i) which mesh hole size provided the most effective barrier to GPA; (ii) which morph of adult aphids (apterous or alate) and/or their progeny could breach the mesh crop cover; (iii) would leaves touching the underside of the cover, as opposed to having a gap between leaf and the mesh, increase the number of aphids breaching the mesh; and (iv) could adults feed on leaves touching the cover by putting only their heads and/or stylets through it? No adult aphids, either alate or apterous, penetrated the mesh crop cover; only nymphs did this, the majority being the progeny of alate adults. Nymphs of the smaller alatae aphids penetrated the three coarsest mesh sizes; nymphs of the larger apterae penetrated the two coarsest sizes, but no nymphs penetrated the smallest mesh size. There was no statistical difference in the number of aphids breaching the mesh crop cover when the leaflets touched its underside compared to when there was a gap between leaf and mesh crop cover. Adults did not feed through the mesh crop cover, though they may have been able to sense the potato leaflet using visual and/or olfactory cues and produce nymphs as a result. As these covers are highly effective for managing TPP on field potatoes, modifications of this protocol are required to make it effective against aphids as well as TPP.

# INTRODUCTION

Potato (*Solanum tuberosum* L.) is a globally important food crop, having the fourth highest production level of 388 million tonnes in 2017 after maize (*Zea mays* L.), wheat (*Triticum aestivum* L.), and rice (*Oryza sativa* L.) (*FAOSTAT, 2020*). In New Zealand, potatoes are the highest grossing vegetable, with consumers purchasing approximately NZ $119 million in 2013 (*Vegetables.co.nz, 2020*). However, potato production is threatened by the tomato potato psyllid (TPP, *Bactericera cockerelli* (Ŝulc 1908) (Hemiptera, Triozidae)). This phloem-feeding insect originated in North and central America and was first identified in New Zealand in 2006 (*Teulon et al., 2009*) and in Western Australia in 2017 (*International Plant Protection Convention, 2017*). TPP feeds on plants in the Solanaceae and Convolvulaceae families (*Wallis, 1955*), and can cause complete crop loss (*Munyaneza, 2013*; *Munyaneza, 2014*).

Due to the negative impacts of TPP on potatoes in New Zealand, organic farmers asked researchers to investigate non-chemical management approaches. Biological options such as the predatory mite *Anystis baccarum* L. (*Geary et al., 2016*) were explored, but without commercial success. In the laboratory, the coccinellid beetle *Cleobora mellyi* (Mulsant 1850) has very high consumption rates of TPP, but this remains to be confirmed in glasshouse and field crops (*O'Connell et al., 2012*; *Pugh, O'Connell & Wratten, 2015*).

'Mesh crop covers' (also referred to as 'insect nets', 'Agronets', 'insect exclusion screens') have been used in Europe for many years to protect a wide range of crops, but not potatoes, from both invertebrate and vertebrate pests (*Collier, 2001*; *Collier, 2002*; *Hill, 1987*; *Saidi et al., 2013*). In New Zealand, however, such covers have only recently been investigated experimentally. Mesh crop covers with a hole size up to 0.6 mm were completely impermeable to TPP in laboratory tests, and were able to reduce TPP populations in field trials to very low levels, even outperforming insecticides (*Merfield, 2017*; *Merfield et al., 2015a*; *Merfield, Hale & Hodge, 2015b*; *Merfield et al., 2019*).

Despite the promising results obtained with mesh crop covers for control of TPP, an unexpected result was that aphids, believed to be mostly the green peach aphid (GPA, *Myzus persicae* (Sulzer, 1776) (Hemiptera: Aphididae)) appeared in large numbers under the mesh crop covers, particularly in the 2016–17 field trials where their populations were significantly higher than uncovered controls, even though the edges of the mesh crop covers were dug into the soil, creating a complete seal (*Merfield, 2017*; *Merfield et al., 2019*).

Aphids can significantly affect plant growth and development, reducing yields. In addition, GPA in particular, is a vector of many plant viruses that also cause significant yield losses (*Capinera, 2001*).

The GPA is the most common and widespread aphid on potatoes in New Zealand, as it feeds on many plant species (*Stufkens & Teulon, 2001*). It is also the most economically important aphid on potatoes, both in New Zealand and worldwide, because it transmits both potato virus Y and leaflet curl virus, which are among the most damaging of the potato viruses (*Marczewski, 2001*; *Saguez et al., 2013*; *Srinivasan et al., 2013*; *Woodford, 1992*). The main management tool for aphids in potatoes is insecticides. However, the GPA has

developed resistance to a number of these (*Bass et al., 2014*; *Foster, Devine & Devonshire, 2007*), which means that both novel and improved non-chemical control techniques are required.

With mesh crop covers being highly effective for TPP management on potatoes, the major challenge is understanding how aphids are circumventing mesh crop covers. Also it is unclear if adults can, from outside the mesh crop cover, feed on potato leaves touching the underside of it. If so, this means aphids outside the mesh crop cover could transmit viruses to potatoes under it. This could discourage seed-potato growers from using the mesh crop cover as a management option for TPP because of virus transmission to tubers intended for propagation.

With these gaps in knowledge, the present research was therefore designed to investigate (i) if aphid stylets can penetrate mesh crop cover of different hole sizes; (ii) if there is a difference in this respect between alate and apterous adults and/or if their progeny have the ability to penetrate mesh crop cover as individuals; (iii) if having potato leaves touching the mesh crop cover from below increases the number of individual aphids penetrating it, compared to when the leaves do not touch the mesh crop cover; and (iv) if adult aphids are capable of feeding on potato leaves through the mesh crop cover.

## MATERIALS AND METHODS

GPA was sourced from a colony cultured on *Brassica rapa* subsp. *chinensis* (L.) pak choi (cultivar: Mei Qing Choi F1) kept at Lincoln University in a controlled-temperature room. The room was kept at 16 h day length, temperature of 23 °C with a 4 °C range and 60% relative humidity. Potato plants (cv. Ilam Hardy) were grown in a glasshouse at the Lincoln University plant nursery to provide leaflets for the laboratory experiment.

For the laboratory work, two, nine-cm diameter Petri dishes were used to create two compartments separated by mesh crop cover; the top dish contained the aphids and the bottom one a single potato leaflet. In the 'control' treatments, three aphids were placed in the bottom dish (Fig. 1). A piece of moist tissue paper was placed in the bottom dish to maintain humidity. Leaflets were then collected from potato plants and cotton wool was wrapped around the petiole of the leaflet, which was inserted into an Eppendorf tube filled with water to maintain leaflet turgidity. The tube with leaflet inserted was placed in the lower dish with the adaxial surface facing up. The mesh crop cover was carefully glued around the full circumference of the opening between the two dishes, because, in previous experiments by *Hodge, Bluon & Merfield (2019)*, aphids could locate and penetrate minimal gaps between the mesh crop cover and hard surfaces. The two Petri dishes were then held together with plastic food wrap. For the mesh crop cover treatments, three adult aphids were inserted through a hole (15 mm diameter) in the top of the upper dish, after which the hole was sealed with mesh crop cover ($0.15 \times 0.15$ mm mesh) held in place by adhesive tape.

There was a total of 24 treatments in a $4 \times 2 \times 3$ factorial design: four mesh crop cover hole sizes × two aphid morphs (apterous or alate) × (three leaflet/aphid positions), in a randomised complete block, with five replicates. Eight of these treatments were controls,

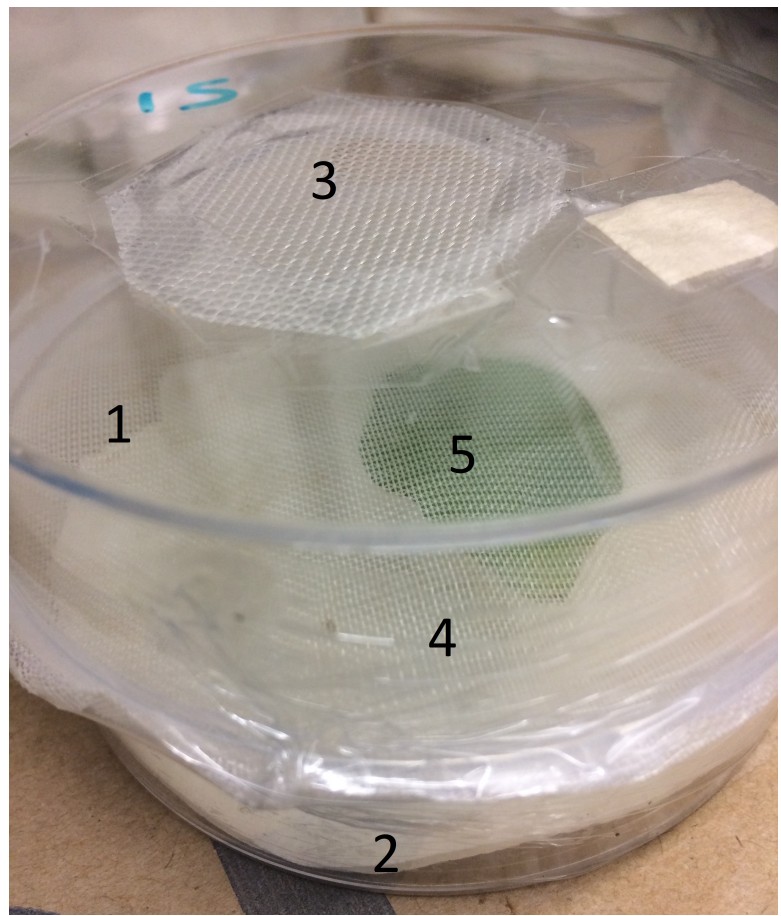

**Figure 1** Experimental set up: (1) top Petri dish, (2) bottom Petri dish, (3) hole for aphid introduction, (4) mesh between dishes, (5) Eppendorf tube and leaflet. For the eight 'control' treatments, aphids were placed below the mesh, while for the other 16 treatments, aphids were placed above the mesh.

one for each of the 4 mesh crop cover hole sizes and 2 aphid morphs, in which three aphids were placed directly on the leaflet under the mesh crop cover. The other 16 (= $4 \times 2 \times 2$) treatments had a mesh crop cover barrier between aphids and leaflets. The mesh crop cover hole sizes stated by the manufacturer were 0. $15 \times 0.15$ mm, $0.15 \times 0.35$ mm, $0.3 \times 0.3$ mm and $0.6 \times 0.6$ mm. Aphid morphs were apterous or alate, while leaflet positions were touching or not touching the mesh crop cover barrier. The experiment ran for 72 h, at which point the number of aphids, both adults and nymphs, on the leaflets were counted. The experiment was conducted in a controlled temperature room as above.

To investigate whether adult aphids could feed through the mesh crop cover, all adults in the upper dish (size 00) at 12, 24, 36 and 48 h through the opening of the top Petri dish. Therefore, 240 individuals were touched four times each, giving a total of 960 tests of feeding. Those that moved following probing were considered to not have been feeding, while those that remained in the same position following probing were judged as having

**Table 1  Measurements of each mesh type.**

| | Mesh size (mm) | | | | | | | |
| | 0.15 × 0.15 | | 0.15 × 0.35 | | 0.3 × 0.3 | | 0.6 × 0.6 | |
|---|---|---|---|---|---|---|---|---|
| Mean | 0.14 | 0.13 | 0.15 | 0.32 | 0.37 | 0.23 | 0.52 | 0.52 |
| Max. | 0.16 | 0.15 | 0.18 | 0.37 | 0.53 | 0.31 | 0.57 | 0.55 |
| Min. | 0.12 | 0.10 | 0.12 | 0.30 | 0.27 | 0.17 | 0.45 | 0.47 |

their stylets inserted into the leaflet and therefore, to be feeding (*Auclair, 1963*; *Giordanengo et al., 2010*).

Mesh crop covers with hole sizes 0.15 × 0.15 mm and 0.15 × 0.35 mm were supplied by AB Ludvig Svensson (http://www.ludvigsvensson.com) as ECONET 1515 and ECONET 1535. Those measuring 0.3 × 0.3 mm and 0.6 × 0.6 mm were supplied by Crop Solutions Ltd. (http://www.cropsolutions.co.uk) and were custom-made for an earlier field trial (*Merfield, 2017*). To test the accuracy of the measurement for each mesh crop cover hole size used in this experiment, ten random samples of each type were selected and 10 holes of each sample were measured under a Nikon SMZ25 microscope (magnification range 0.63–15.75×). The mean, minimum and maximum mesh hole size measurements are presented in Table 1.

All data were analysed in a randomised block analysis of variance (ANOVA) (with a factorial treatment structure) using GenStat$^{®}$ 18th edition. The response variable (number of aphids on potato leaflets) was subjected to a square-root transformation to normalise the data before analysis. Also, the analysis was split into two ANOVAs to achieve homogeneity of variance. (1) The eight controls (4 mesh hole-sizes ×2 aphid-morphs), which were relatively highly variable, were analysed separately as a 4× 2 factorial with 5 blocks. (2) For the 16 non-control treatments, six were all zeroes and hence had zero variability, so were omitted from the analysis, leaving 10 treatments which were analysed as a $(2 \times 2 + 1) \times 2$ factorial with 5 blocks.

# RESULTS

For the eight control treatments, with aphids below the mesh, there were no significant differences in nymph numbers produced by the two aphid morphs, no significant linear or quadratic components of mesh crop cover hole size (assuming these were in the ratio 1: 2: 3: 6), nor any significant interaction components (Table 2).

In the 16 treatments with adults placed above the mesh cover, only nymphal, not adult aphids were able to pass through the mesh. Nymphs of the smaller alate adults penetrated the 0.15 × 0.35, 0.3 × 0.3 and 0.6 × 0.6 meshes but not the 0.15 × 0.15 mesh (Table 2). Nymphs of the larger apterous adults penetrated the 0.3× 0.3 and 0.6× 0.6 meshes but did not breach the 0.15× 0.15 and 0.15× 0.35 ones.

Comparing when leaflets touched or did not touch the mesh crop cover, there was no significant difference in the number of nymphs penetrating the mesh ($P = 0.612$). The interaction between aphid morph (alate and apterous) and the leaflet touching mesh crop cover or not was also not significant ($P = 0.066$).

**Table 2** Mean ($\sqrt{\,}$) number of aphid nymphs of apterous and alate parents breaching different mesh sizes when leaflets were touching the mesh or not. (A) Controls were statistically analysed. (B) Treatments with means in brackets indicate those that were omitted because they had zero mean and zero variance. m.e. = main effect.

| | Aphid morph | Mesh size (mm) | Mean of square root of # of nymphs below mesh | Back transformed means |
|---|---|---|---|---|
| **(A)** | | | | |
| Control | Apterous | 0. 15×0.15 | 2.460 | 6.05 |
| | | 0. 15×0.35 | 2.202 | 4.85 |
| | | 0. 3×0.3 | 1.940 | 3.76 |
| | | 0. 6×0.6 | 2.448 | 5.99 |
| | Alate | 0. 15×0.15 | 1.823 | 3.32 |
| | | 0. 15×0.35 | 1.673 | 2.80 |
| | | 0. 3×0.3 | 2.455 | 6.03 |
| | | 0. 6×0.6 | 1.756 | 3.08 |
| **LSD 5%** | | | **1.235** | |
| Significance of m.e., apterous vs alate | | | not sig. | |
| **(B)** | | | | |
| Leaflet not touching mesh | Apterous | 0.15× 0.15 | (0.000) | 0.00 |
| | | 0.15× 0.35 | (0.000) | 0.00 |
| | | 0.3× 0.3 | 0.483 | 0.23 |
| | | 0.6× 0.6 | 0.883 | 0.78 |
| | Alate | 0.15× 0.15 | (0.000) | 0.00 |
| | | 0.15× 0.35 | 0.200 | 0.04 |
| | | 0.3× 0.3 | 0.283 | 0.08 |
| | | 0.6× 0.6 | 0.829 | 0.69 |
| Leaflet touching mesh | Apterous | 0.15× 0.15 | (0.000) | 0.00 |
| | | 0.15× 0.35 | (0.000) | 0.00 |
| | | 0.3× 0.3 | 0.200 | 0.04 |
| | | 0.6× 0.6 | 0.400 | 0.16 |
| | Alate | 0.15× 0.15 | (0.000) | 0.00 |
| | | 0.15× 0.35 | 0.546 | 0.30 |
| | | 0.3× 0.3 | 1.012 | 1.02 |
| | | 0.6× 0.6 | 1.029 | 1.06 |
| **LSD 5%** | | | **0.905** | |

No aphids, at any time fed through the mesh crop cover, as all moved when probed with the artist's brush.

# DISCUSSION

## Mesh crop cover hole size and aphid penetration

The fact that no adult aphids penetrated the covers, differs from the result of *Hodge, Bluon & Merfield (2019)* studying GPA and the melon aphid (*Aphis gossypii* (Glover 1877)). They found that adult GPA did penetrate LS ECONET 1515, but no GPA adults penetrated the larger hole LS ECONET 1,535, a result which appears inconsistent. *Hodge, Bluon &*

*Merfield (2019)* did, however, find that nymphs penetrated the mesh, in agreement with these results.

That adults did not penetrate mesh, but newly hatched nymphs did, suggesting a means by which aphids entered all mesh covered plots in the field trial of *Merfield (2017)*: adult aphids that alighted on the upper surface of the mesh, produced nymphs, which are small enough to penetrate the mesh, and once through that they can reproduce on the potato crop. This result needs to be verified under field conditions.

The extent to which aphids penetrate mesh and for mesh to control aphids is clearly variable. For example, *Bethke, Redak & Paine (1994)* who worked with adult aphids, found GPA could not pass through a square, brass mesh with hole sizes ranging from 0.19 mm$^2$ to 0.94 mm$^2$, but melon aphid could penetrate meshes larger than 0.46 mm$^2$. Using mosquito nets of knitted polyester, 30 gm$^{-2}$ and a mesh size of 25 holes per cm$^2$, untreated or treated with deltamethrin, supported 50 cm above seedling cabbages (*Brassica oleracea* L.), *Martin et al. (2006)* found the treated nets had significantly lower numbers of plants infested with the mustard aphid (*Lipaphis erysimi* (Kaltenbach 1843)). In subsequent work, *Martin et al. (2013)* used mesh with 0.9 mm diameter, treated with alphacypermethrin and two untreated nets, one with the same hole size and the other with a smaller one of 0.4 mm, and control without mesh, to cover pot-grown cabbages in the field with the mesh supported 30 cm above the plants. They found no cabbage aphid (*Brevicoryne brassicae* L.) under the 0.9 mm alphacypermethrin treated and untreated 0.4 mm mesh, but a small, yet statistically significant greater number, under the untreated 0.9 mm mesh. In comparison, both GPA and mustard aphids (counted together as 'green aphids'), were not present under the treated 0.9 mm mesh but were present in much larger numbers than cabbage aphid in the two other meshes. The untreated 0.9 mm mesh had a statistically significant population 5.8 times larger than the non-covered control while the 0.4 mm mesh had more than twice the number of aphids than the control, although the difference was not statistically significant. *Licciardi et al. (2007)*, using deltamethrin-treated polyester mesh with 40 holes/cm$^2$, kept 50 cm above cabbages, with the mesh either removed during the day ('temporary') or kept permanently on the crop, found that aphids infested the untreated control immediately on planting out the crop. The temporary mesh was infested at 20 days after planting (DAP), and the permanent mesh at 30 DAP, but then, the aphid numbers under the permanent screen rapidly increased to finish at approximately four times that under the temporary mesh. In orchard trials in France and Italy using mesh covers with hole sizes of either 2.2 × 5.4 mm or 3 × 7.4 mm to control codling moth (*Cydia pomonella* L.), *Alaphilippe et al. (2016)* found that rosy apple aphid (RAA, *Dysaphis plantaginea* (Passerini 1860)) and woolly apple aphid (*Eriosoma lanigerum* (Hausmann, 1802)) populations increased. In contrast, *Dib, Sauphanor & Capowiez (2010)* found mesh covers reduced RAA populations and also those of their ant mutualists as well as their predators under covers also used for codling moth control. In a comparison of nets treated and untreated with insecticides against black bean aphids (*Aphis fabae* (Scopoli 1763)) on French bean (*Phaseolus vulgaris* L.), *Gogo et al. (2014)* found both mesh types significantly reduced aphid populations compared with the control, but did not eliminate them.
In a few cases, mesh crop covers appeared to be impenetrable to aphids, or at least can prevent aphids reaching crops, though this varies with aphid species e.g., cabbage aphid cv. GPA (*Martin et al., 2013*), and could be related to size, though other factors, e.g., behavioural, may also have a role. However, in most cases discussed above, mesh covers appeared only to delay aphids reaching the crops, and, in some cases, aphid populations then rapidly built to levels much higher than those in untreated controls or insecticide treatments. It is hypothesised that the mesh is preventing, or at least reducing the range and number of aphid natural enemies that can reach the crop.

Aphids are also not the only insect species that can circumvent mesh. *Talekar, Su & Lin (2003)* and *Licciardi et al. (2007)* reported that adult female moths of the armyworm (*Spodoptera littoralis* (Boisduval 1833)) laid eggs on the top of a hooped screen, and the newly hatched larvae were sufficiently small to penetrate the mesh and drop onto the crop below. *Merfield (2017)* found a large number of lacewings (*Micromus tasmaniae* (Walker 1860)) in plots of potatoes under mesh, following aphid outbreaks under the mesh. Lacewing adults are much too large to penetrate mesh, and the mesh was dug into the soil preventing entry from the edges, so it was hypothesised that the lacewings laid eggs on the mesh, and then the neonate larvae penetrated the mesh. As aphids, lacewings and armyworms would not be expected to lay eggs or produce nymphs in the absence of suitable hosts, this indicates that the insects can detect their hosts through the mesh, even when the mesh is at some distance from the crop (*Licciardi et al., 2007*; *Talekar, Su & Lin, 2003*).

## Difference between the progeny of alate and apterous adults to penetrate mesh crop cover

In the present study, nymphs produced by alate adults penetrated the 0. 15×0.35 mesh crop cover, while those produced by apterous adults did not. This supports the findings of (*Dixon & Wratten, 1971*) who found that alate aphids were smaller, and produced fewer and smaller nymphs than did apterous aphids, which indicates that only the smallest size mesh crop cover hole size (0. 15×0.15 mm) would be an effective barrier. For mesh crop cover with larger hole sizes, neonate nymphs of both adult morphs can enter the mesh crop cover, so both alate and apterous adults are potential threats to the crop. However, apterous aphids, lacking wings, could arrive only on the outside of mesh crop cover from other plants bordering the mesh crop cover. In the field trial of *Merfield (2017)* and *Merfield et al. (2019)*, the periphery of mesh crop cover was kept free of vegetation with residual herbicides and tillage such that apterous aphids walking onto the mesh crop cover should have been eliminated, yet all mesh crop cover plots were infested by aphids. It is therefore believed, that nymphs, which infested the mesh crop cover treatments in those trials, were produced by alate adults.

## Adult aphid feeding through mesh and implications for virus transmission

That adults were not found to feed through the mesh indicates potato viruses will not be transmitted through mesh. However, this result is not direct evidence for lack of transmission, as it only demonstrated a lack of feeding but not a lack of probing. GPA
can also transmit viruses by probing (*Radcliffe & Ragsdale, 2002*). Direct testing of virus transmission is required, using virus-infected aphids to test the rates of transmission when mesh is present or not.

## Solutions for aphids penetrating mesh crop covers

That some aphid species, and also armyworm (*Licciardi et al., 2007*; *Talekar, Su & Lin, 2003*) can circumvent mesh indicates that for some insect pests, additional control measures will be required when using covers. Three general approaches are suggested: (i) modify the physical properties of the mesh to prevent pests alighting or reproducing on it; (ii) use insecticide or repellent-treated mesh covers to kill or repel alighting pests; (iii) use supplemental biological control. For aphids on potatoes, the approach used should ideally also ensure that virus transmission is minimised.

A range of research has altered the physical properties of mesh crop covers to both repel and attract insects. Some of this research can be considered for field potatoes. These alterations include: different colours (*Ben-Yakir et al., 2012a*), and ultra-violet light properties (*Ben-Yakir et al., 2012b*), such that the mesh prevents aphids alighting.

Insecticide treated mesh crop covers have been discussed above, and, are clearly highly effective in some situations. A further refinement of this approach has been to create attractive areas on the mesh, e.g., yellow spots, with only the attractive areas treated with insecticides (*Ben-Yakir et al., 2014*). Chemical repellants can also be used in place of biocides (*Martin et al., 2013*). However, the insecticides and repellents are lost from the mesh covers over time, so the efficacy of treated mesh covers reduces, meaning their effective lifespan can be less than that of untreated mesh covers (*Dáder et al., 2015*). Treating mesh also increases cost, and may require personnel to use protective equipment when handling and storing the mesh covers. If a single chemical is used to treat the mesh, this is likely to create increased pressure for evolved resistance, both against biocides, and, also repellents (*Foster, Devine & Devonshire, 2007*). This could be partially mitigated through impregnating the mesh covers with multiple insecticides with different modes of action, but, at further increased cost.

A range of biological control approaches could be used. In the case of armyworm (discussed above), infestations were effectively controlled using the biological control agent (BCA) *Bacillus thuringiensis* (*Talekar, Su & Lin, 2003*). In field crops, microbial BCAs could be applied through the mesh (*Ester, Van de Zande & Frost, 1994*; *Merfield, 2018*). Augmentative biological control using predators and parasitoids is widely used in protected cropping, such as glasshouses. Mesh crop covers are considered to be a form of protected cropping, in that they should prevent introduced BCAs escaping from the crop, so existing glasshouse biocontrol solutions could be leveraged. For example, *Gagnon Lupien, Aoun & Chouinard (2014)* suggested the release of the commercially available coccinellid *Adalia bipunctata* (L.) to compensate for the lack of aphid natural enemies under mesh crop covers. Other biological control agents (BCAs) that have been used to successfully manage GPA in protected envrionments include, *Aphidius matricariae*, *Aphidius colemani* (*Zamani et al., 2007*) and *Micromus tasmaniae* (*Harcourt, 1996*; *Jonsson et al., 2009*). Augmentative biological control with insects could be further enhanced by the use of banker plants

(which provide supplemental food for beneficial insects, which helps them maintain a stable population) (*Frank, 2010*) grown under the mesh / among the potato crop; this approach is already used for control of GPA (*Andorno & López, 2014*).

## CONCLUSIONS

Because of commercial availability and cost, it is recommended that the 0. 6×0.6 mm mesh crop cover be used to manage TPP on potatoes, even though this mesh can be penetrated by GPA nymphs. Further research is required to confirm GPA nymphs' ability to penetrate mesh covers in the field and whether or not mesh is an effective barrier to virus spread, both by adults and nymphs. A considerable amount of research is required into methods to control GPA and other aphid species in mesh-covered potato crops. If mesh crop cover is combined with additional approaches that achieve a high level of GPA control and therefore that of potato viruses, the benefits to the potato seed industry are likely to be considerable. Mesh crop covers have been in use across Israel and Europe since the early 1990s and the area used in Europe is estimated at 50,000 ha, with hundreds of ha on individual farms (Ian Campbell, Crop Solutions Ltd., personal communication). Mesh crop covers are therefore a completely farm-proven, large-scale, technology. If an effective means of managing aphids on potatoes under mesh crop covers can be developed, it has the potential to be a viable option for management of TPP on potato in New Zealand and globally.

## ACKNOWLEDGEMENTS

The authors thank Brent Richards and Leona Meachen from Lincoln University plant nursery for assisting in growing potato plants; Jenny Brookes for the supply of GPA; Brian Kwan for technical advice; Janine Johnson for editorial assistance; Crop Solutions Ltd and AB Ludvig Svensson for donating mesh crop cover samples.

### Funding

This work was supported by the Ministry of Foreign Affairs and Trade, New Zealand, and the Bio-Protection Research Centre, Lincoln University, New Zealand. The funders had no role in study design, data collection and analysis, decision to publish, or preparation of the manuscript.

### Grant Disclosures

The following grant information was disclosed by the authors:
Ministry of Foreign Affairs and Trade, New Zealand.
Bio-Protection Research Centre, Lincoln University, New Zealand.

### Competing Interests

David J. Saville is employed by Saville Statistical Consulting Limited and is contracted to the Bio-Protection Research Centre, Lincoln University, for statistical advice. Stephen

D Wratten is an Academic Editor for PeerJ. The authors declare there are no competing interests.

## Author Contributions

- Howard London conceived and designed the experiments, performed the experiments, prepared figures and/or tables, authored or reviewed drafts of the paper, and approved the final draft.
- David J. Saville conceived and designed the experiments, analyzed the data, authored or reviewed drafts of the paper, and approved the final draft.
- Charles N. Merfield and Stephen D. Wratten conceived and designed the experiments, authored or reviewed drafts of the paper, and approved the final draft.
- Oluwashola Olaniyan conceived and designed the experiments, performed the experiments, authored or reviewed drafts of the paper, and approved the final draft.

## Data Availability

The raw data is available as a Supplementary File.

## Supplemental Information

Supplemental information for this article can be found online at http://dx.doi.org/10.7717/peerj.9317#supplemental-information.

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
