# Peer review of "The ability of the green peach aphid (Myzus persicae) to penetrate mesh crop covers used to protect potato crops against tomato potato psyllid (Bactericera cockerelli)"

_PeerJ, doi:10.7717/peerj.9317_

## Round 0.1 · original submission · Major Revisions

· Academic Editor

Major Revisions

This is a small lab test that has been fleshed out with a high amount of speculative information. The value really is in the determination that some aphid nymphs may penetrate screen sizes that were shown to stop adult aphids.

The title (and text) indicates the screens stop psyllids, and the problem is aphids. Psyllid control with row covers was not tested in this study. The study that they cite regarding psyllid control is rather weak. Row covers were tested for two years, but each year had a different experimental design so replication was lacking. No measure of CLas transmission was collected, so the economic value and effectiveness of row covers was not proven.

One of the reviewers, with substantive expertise in the area requires substantive more thought on the actual research carried out: e.g. Munyaneza has shown that psyllids can be stopped with row covers (set up as 'hoop cages) and zebra chip transmission can be eliminated. However, if set up properly, row covers can also prevent aphid transmission of pathogens. The Merfield study cited by the authors had the covers simply lying on the plants. This led to more speculation regarding how the results might change if the row covers were set up differently. Given that this was a lab test with aphids, the speculation on how to improve psyllid control out of place.

Reviewer 1 ·

Basic reporting

The study presented in this manuscript is a relatively small yet straightforward lab test. However, there are some critical assumptions and a great deal of speculation included. For example, this study did not test the screens against TPP, but the authors did cite a study by Merfield et al. (2015). Interestingly, the Merfield study found TPP nymphs and adults underneath the row covers (see Fig 1 in Merfield et al.). Thus, while TPP populations may have been reduced under the row covers, the impact on occurrence of Clas did not appear to be measured. Thus, the assumption that mesh covers will provide economic levels of CLas reduction may be premature. In addition, the authors used considerable space in the manuscript to describe how biocontrol agents might be used in conjunction with the screening to control GPA in the field. However, no experiments were conducted to test their hypothesis that biocontrol agents can provide economic suppression under row covers in the field, so this discussion could be shortened. In addition, it was not clear to me why adding biocontrol agents were the only options discussed. Could the screens be treated with pesticides or an aphid repellent and achieve the same goal at less expense? How about coloured screening?

The list of locations provided in the first sentence of the abstract suggests this is the range of the pest. TPP is a pest in many other countries (Butler and Trumble 2012. Terrestrial Arthropod Reviews 5: 87–111). I suggest the authors consider stating the pest is a problem on potatoes throughout most of Central America and all of North America, as well as New Zealand.

It would be useful if the authors compared their work to other studies where screens with different mesh sizes were tested against green peach aphids. For example, Bethke et al. 1994. (Screens deny specific pests entry to greenhouses. California Agriculture. California Agriculture 48(3): 37-40) indicated the mesh sizes that prevented entry to green peach aphids. Do their results agree with what was found in this study? Is other literature available?

Some of the potato fields in NZ are quite large. Is covering all of the rows with screening economically viable? Or is this screening primarily being considered for seed potatoes?

Finally, the authors repeatedly point out that the interaction between aphid morph (alate and apterous) and the leaflet touching mesh or not was ‘nearly significant’. They then indicate several times why this so important. Finding importance or trends in non-significant data is questionable, and almost never justified.

Minor stuff:

Line 193: add ‘aphids’ after “apterous”.

In the Munyaneza references, the genus and species name of the TPP need to be in italics.

Experimental design

The experimental design is OK. The statistics appear to have been conducted appropriately, but see the comment above on "nearly significant" findings.

Validity of the findings

see above

Reviewer 2 ·

Basic reporting

The authors evaluated the effectiveness of using insect-proof mesh screens as a barrier for green peach aphid, Myzus persicae and studied the effect of different mesh hole size, the stage of the green peach aphid that penetrates the mesh, to find whether or not leaves touching the underside of the mesh would increase insect feeding, and if adults can feed from outside of the mesh.
The manuscript is very well written with well laid hypotheses and good flow of information in the introduction. The authors have taken effort to consider various factors that would possibly affect the outcome of the study and taken necessary precautions at various stages of the study right from formulating the hypotheses to obtaining their final results keeping the reader informed.

Experimental design

Given the various factors involved in the study, the authors chose to do a factorial design. However, I find the authors have also mentioned about the randomized complete block design in line 133 and wish the authors explained about the 'so eight "controls"'.
Lines 127 to 131 in the methods sections is not very clear and may be rewritten for clarity.

Validity of the findings

The authors have done a great job reporting their findings. The findings are novel and informative. The authors also mentioned introducing bio-control agents under the mesh as a future scope of research.
In lines 184 and 185 the authors tried to explain a non-significant result. If results are non-significant there is no need to mention if the actual differences were small are large.

Additional comments

Some minor corrections are mentioned which do not detract the quality of the manuscript in any way.
Line 44 - delete 'the' glo
Line 52 - use quotation marks for the scientific name of the pathogen
Line 63 - delete 's' from covers and insert "to" reduce
Line 102 - potatoes under "the mesh cover"
Line 123 - cotton wool was "wrapped around" the petiole
Line 174 - large apterous "adults" ??
Line 201 - disperse "by" wind currents

The discussion is a little deviating presumably out of the scope of the listed objectives as to what mechanisms/cues the aphids adopt to land on the food source.
I have not checked the citations. So it may be good to verify the reference section.
Overall, the authors have done a great job presenting their research in a scientific manner that is acceptable to the standards of PeerJ. I therefore recommend publication of this manuscript in PerrJ following minor corrections.

Reviewer 3 ·

Basic reporting

This paper details the use of Mesh crop covers on potatoes to protect against potato psyllids and additional challenges posed by aphids. They examine the different mesh sizes to manage the psyllid in potato. They further examined green peach aphid ability to penetrate different mesh hole sizes. Overall, this study is nicely done, and the methods seem sound. The analysis seems appropriate. The manuscript is well written with no revisions required.

Experimental design

Research methods were seem sound. The analysis was appropriate. The manuscript is well written with no revisions required.

Validity of the findings

The findings were well stated.

---

## Round 0.2 · accepted · Accept

· Academic Editor

Accept

The authors and reviewers have done a great job in increasing the readability and validity of the study carried out here. I am happy that the review process has added value to this manuscript, and has taken it to the level ready for publication.

Reviewer 1 ·

Basic reporting

The revised manuscript has now included appropriate comparisons with previous literature.

Experimental design

no comment

Validity of the findings

This aspect of the manuscript is much improved. The authors have reduced the speculation and focused the manuscript on what was accomplished.

Additional comments

This version of the manuscript is much improved. The authors have adequately addressed all of my concerns, and I can recommend acceptance.